# Association of Fabry Disease with Hearing Loss, Tinnitus, and Sudden Hearing Loss: A Nationwide Population-Based Study

**DOI:** 10.3390/jcm11247396

**Published:** 2022-12-13

**Authors:** Yen-Fu Cheng, Sudha Xirasagar, Chin-Shyan Chen, Dau-Ming Niu, Herng-Ching Lin

**Affiliations:** 1Department of Medical Research, Taipei Veterans General Hospital, Taipei 112, Taiwan; 2Department of Otolaryngology-Head and Neck Surgery, Taipei Veterans General Hospital, Taipei 112, Taiwan; 3Department of Otolaryngology-Head and Neck Surgery, School of Medicine, National Yang Ming Chiao Tung University, Taipei 112, Taiwan; 4Institute of Brain Science, National Yang Ming Chiao Tung University, Taipei 112, Taiwan; 5Research Center of Sleep Medicine, College of Medicine, Taipei Medical University, Taipei 110, Taiwan; 6Department of Health Services Policy and Management, Arnold School of Public Health, University of South Carolina, Columbia, SC 29208, USA; 7Department of Economics, National Taipei University, 151 University Rd. San Xia, New Taipei City 237, Taiwan; 8Department of Pediatrics, Taipei Veterans General Hospital, Taipei 112, Taiwan; 9Institute of Clinical Medicine, National Yang Ming Chiao Tung University, Taipei 112, Taiwan; 10Sleep Research Center, Taipei Medical University Hospital, Taipei 110, Taiwan; 11School of Health Care Administration, Taipei Medical University, Taipei 110, Taiwan

**Keywords:** Fabry disease, epidemiology, hearing loss, sudden deafness

## Abstract

Hearing loss and the related otologic manifestations are receiving increased scrutiny as significant causes of morbidity in Fabry disease. However, the relative risks of auditory deficits among patients with Fabry disease relative to the general population without a diagnosis of Fabry disease have not been studied. This study aims to explore the associations between Fabry disease and hearing-related manifestations using a nationwide population-based dataset. We identified study patients for this cross-sectional study from the 2015–2017 claims databases of the Taiwan Longitudinal Health Insurance Database 2005. We first identified 2312 patients aged over 20 years with a diagnosis of Fabry disease. We used propensity score matching to select five comparison patients per patient with Fabry disease and 11,560 comparison patients without Fabry disease. We used multivariable logistic regressions to estimate the odds ratios (ORs) and corresponding 95% confidence intervals (CIs) for tinnitus, hearing loss, and sudden deafness among Fabry disease patients vs. comparison patients. Chi-square tests showed statistically significant differences between patients with and without Fabry disease in the prevalence rates of tinnitus (16.7% vs. 11.7%, *p* < 0.001), hearing loss (7.5% vs. 6.2%, *p* = 0.014) and sudden deafness (1.7% vs. 1.0%, *p* = 0.005). Multiple logistic regression revealed that patients with Fabry disease were more likely to suffer from tinnitus, hearing loss and sudden deafness, with adjusted odds ratios of 1.513 (95% CI = 1.336–1.713), 1.246 (95% CI = 1.047–1.483), and 1.681 (95% CI = 1.166–2.423), respectively. We found that Fabry disease is significantly associated with certain auditory manifestations, including hearing loss, sudden deafness, and tinnitus.

## 1. Introduction

Fabry disease, also known as Anderson-Fabry disease, is an X-linked inherited lysosomal storage disorder first described by Drs. William Anderson and Johanness Fabry independently in 1898 [1,2,3]. Fabry disease has been found to be an underdiagnosed condition and can occur in all ethnic groups and geographic areas, affecting approximately 1:14,000 to 1:117,000 people worldwide [1,2,3,4], although recent newborn screening studies have suggested a higher incidence [5,6,7,8,9,10,11,12]. The incidence is 1:4600 to 1:7700 in Japan, while it is reported to be as high as 1:600 to 1:1250 in Taiwan [5,6,7,8,9,10].

Fabry disease is caused by mutations in the enzyme protein-coding gene GLA (galactosidase alpha) mapped to the X-chromosome (Xq21.3-q22). Galactosidase alpha catalyzes the removal of terminal α-galactose groups from substrates such as glycoproteins and glycolipids. The loss of functional GLA enzymes leads to the progressive accumulation of glycosphingolipids, primarily globotriaosylceramide (Gb3) and related glycosphingolipids, within the lysosomes of various cell types of multiple organ systems, especially endothelial cells. Clinically, Fabry disease represents a vascular disease and may result in damage to the cardiovascular, renal, dermatologic, and nervous systems of affected patients [13,14,15]. Due to the vague and nonspecific manifestations, Fabry disease is frequently mis- or underdiagnosed, and the average time to diagnosis can be more than one decade [16,17].

Hearing loss and the related otologic manifestations are receiving increased scrutiny as significant causes of morbidity in Fabry disease. Despite nonlethal complications for Fabry disease, otologic manifestations such as hearing loss, sudden deafness, and tinnitus have been featured in some case reports and case series of patients with Fabry disease [18,19,20,21]. However, to the best of our knowledge, the relative risks of the aforementioned auditory deficits among patients with Fabry disease relative to the general population without a diagnosis of Fabry disease have not been studied. Furthermore, due to the shared pathogenesis of vasculopathy and the high prevalence of Fabry disease in the East Asian population [5,6,7,8,9,10], we aimed to explore the association between Fabry disease and hearing-related manifestations using a nationwide population-based dataset.

## 2. Methods

### 2.1. Database

We identified study patients for this cross-sectional study from Taiwan’s Longitudinal Health Insurance Database 2005 (LHID2005). Taiwan initiated the National Health Insurance (NHI) program in 1995, a single-payer, mandatory social insurance program covering all Taiwanese citizens. The LHID2005 comprises the registration files of 2,000,000 NHI beneficiaries and claims files with data on ambulatory care visits made, diagnoses, details of ambulatory care orders, inpatient expenditures by admissions, details of inpatient orders, and details of prescriptions dispensed at NHI-contracted pharmacies. Many researchers from Taiwan universities, research institutes, and medical centers have received access to deidentified datasets provided by the LHID2005 for epidemiological studies of diseases and treatments.

This study was approved by the institutional review board of Taipei Medical University (TMU-JIRB N202203211) and is compliant with the Declaration of Helsinki. Because we used deidentified administrative data, informed consent was waived.

### 2.2. Identification of Study Patients

This cross-sectional study was designed to compare the study group with a comparison group. To identify patients in the study group, we extracted the claims records of all patients aged over 20 years with a diagnosis of Fabry disease (ICD-9-CM code 272.7 or ICD-10-CM code E75.21) in ambulatory care visits to clinics or outpatient departments of hospitals between 1 January 2015 and 31 December 2017. We identified 2312 patients with Fabry disease in the study group. We retrieved a comparison group from the remaining LHID2005 enrollees aged ≥ 20 years from the Registry of beneficiaries. We first excluded all enrollees who had ever received a Fabry disease diagnosis prior to 2015. Thereafter, we used propensity score matching to select comparison patients. We calculated a propensity score for each enrollee with and without an FD diagnosis using selected demographic variables, including age, sex, and monthly income (NT)$0~15,840, NT$15,841~25,000, ≥NT$25,001; the average exchange rate in 2021 was US$1 ≈ NT$28), geographic location (Northern, Central, Southern, and Eastern) and urbanization level of the patient’s residence (five levels, 1 meaning the most urbanized and 5 being the least urbanized). The matching ratio was five comparison patients to one patient with Fabry disease. However, exact score-matched comparison patients may not be found for every patient with Fabry disease. We therefore chose the alternative method of nearest neighbor within calipers to match comparison patients (apriori value for the calipers is +/−0.01). The final study sample included 2312 patients with Fabry disease and 11,560 matched patients without Fabry disease.

### 2.3. Measures of Outcomes

The outcome variables of interest were tinnitus, hearing loss and sudden deafness. We identified these cases by finding the respective ICD diagnosis codes in the ambulatory visit claims, tinnitus (ICD-9-CM code 388.3 or ICD-10-CM codes H93.1, H93.11, H93.12, H93.13 or H93.19), hearing loss (ICD-9-CM code 389 or ICD-10-CM codes H90 or H91) or sudden deafness (ICD-9-CM code 388.2 or ICD-10-CM code H91.2).

### 2.4. Statistical Analysis

Statistical analyses were carried out using the SAS system (SAS System for Windows, vers. 9.4, SAS Institute, Cary, NC, USA). We performed chi-square tests and t tests to examine differences in patient demographics between patients with Fabry disease and comparison patients. We used multivariable logistic regressions to estimate the odds ratios (ORs) and 95% confidence intervals (CIs) for tinnitus, hearing loss, and sudden deafness among Fabry disease patients vs. comparison patients. We used two-sided *p* < 0.05 for statistical significance.

## 3. Results

Table 1 shows the sociodemographic characteristics of patients with and without Fabry disease. There were no statistically significant differences in age (*p* = 0.990), sex (*p* > 0.999), monthly income (*p* = 0.994), geographic location (*p* > 0.999) or residential urbanization level (*p* > 0.999) between patients with and without Fabry disease.

Table 2 presents the prevalence rates of tinnitus, hearing loss and sudden deafness among patients with and without Fabry disease. Among the total sample, the prevalence rates of tinnitus, hearing loss and sudden deafness were 12.6%, 6.4%, and 1.1%, respectively. Chi-square tests indicated statistically significant differences between the groups with and without Fabry disease in the rates of tinnitus (16.7% vs. 11.7%, *p* < 0.001), hearing loss (7.5% vs. 6.2%, *p* = 0.014), and sudden deafness (1.7% vs. 1.0%, *p* = 0.005).

A logistic regression analysis showed that the odds ratios for tinnitus, hearing loss and sudden deafness were 1.502 (95% CI = 1.328–1.699), 1.240 (95% CI = 1.044–1.473), and 1.678 (95% CI = 1.165–2.418), respectively, for patients with Fabry disease relative to comparison patients (Table 3). Multiple logistic regression revealed that after adjusting for age, income, geographic location, and residential urbanization level, patients with Fabry disease were more likely to have suffered tinnitus, hearing loss and sudden deafness. The respective adjusted odds ratios were 1.513 (95% CI = 1.336–1.713), 1.246 (95% CI = 1.047–1.483), and 1.681 (95% CI = 1.166–2.423).

## 4. Discussion

To our knowledge, this may be the first large-scale population-based epidemiological study to explore the risk of hearing loss, sudden deafness, and tinnitus among patients with Fabry disease. Our results show that Fabry disease is significantly associated with certain auditory manifestations, namely hearing loss, sudden deafness, and tinnitus. Using a propensity-score matched case-control study design, we found that the odds of sudden deafness among persons with Fabry disease are 1.68 that of the general population without a diagnosis of Fabry disease, while the odds for tinnitus and hearing loss are 51% and 25% higher, respectively.

The clinical manifestations of Fabry disease are mainly caused by functional defects in the GLA enzyme, which may lead to progressive accumulation of Gb3 in lysosome-containing cells, including the vascular endothelium, smooth muscle cells, and cardiomyocytes [22]. The accumulation of Gb3 in the endothelial cells of the vasculature induces a cascade of pathologic processes, including the induction of the excessive production of reactive oxygen, the release of proinflammatory cytokines, and the activation of the innate inflammatory response, thus leading to the release of prothrombotic factors and subsequent extracellular matrix deposition and fibrosis [23,24].

Vascular pathology is the leading cause of hearing-related disorders, including hearing loss, sudden deafness, and tinnitus. Sudden deafness, also called sudden sensorineural hearing loss, is a medical emergency that warrants an urgent clinical visit and timely management. Sudden deafness is defined as a hearing loss of 30 decibels or more over at least three contiguous frequencies occurring within a period of 72 h or less [25,26]. Several underlying causes of sudden deafness have been proposed, including vascular disorders, microbial infections, autoimmune diseases, and inner ear membrane rupture [27,28,29,30], yet there has been no conclusive evidence to support these hypotheses. However, a vascular etiology has been recently favored in the pathogenesis of sudden deafness in light of certain clinical features, such as the abrupt onset and its high association with some thromboembolic diseases [31,32]. Similarly, vascular pathology has been reported to be the leading cause of progressive sensorineural hearing loss and tinnitus in several experimental and clinical studies [33,34].

The underlying mechanism of hearing loss in Fabry disease patients is not yet understood. The cochlea, the hearing organ embedded in the temporal bone, is highly dependent on the blood and oxygen supply to maintain its normal function [35]. The cochlea is regarded as an end-organ in terms of the vascular supply, as its blood supply relies on a single artery, the labyrinthine artery [33]. Similar to the cases for the heart and kidney [36], this type of single blood supply source renders the cochlea vulnerable to any circulatory inadequacy. In addition to an acute cochlear ischemic event such as sudden deafness, chronic and progressive insufficiency of the cochlear blood supply may cause increased vulnerability to ototoxic events and the accelerated aging of the cochlea, leading to progressive hearing loss and/or tinnitus. The vascular origin of Fabry disease is supported by the observations of Schachern et al. They were the first to describe the cochlear histopathology among patients with Fabry disease suffering from sensorineural hearing loss [37]. They found cytoplasmic vacuolization of the vascular endothelial cells, hair cell loss in the basal turn of the cochlea, spiral ganglion loss, stria vascularis, and spiral ligament atrophy.

Our study may be the first and largest population-based study to explore the risk of hearing disorders, including hearing loss, sudden deafness, and tinnitus, among patients with Fabry disease. A key strength of the study lies in the usage of a nationwide dataset of outpatient utilization claims of an entire nation’s population, curated and stored by the NHIRD for research. The universal health insurance coverage system of Taiwan provides comprehensive healthcare coverage to all citizens (over 99.9% of Taiwan’s resident population). Fabry disease, with a global incidence of 1 in 14,000–117,000 people [1,2,3,4], has a much higher incidence in Taiwan [5,6,7,8]. Through newborn screening and related genetic tests, our study team and other researchers have reported a high prevalence (1 in 600 to 1250 people) of a pathogenic variant of late-onset cardiac Fabry disease (IVS4 + 919 or c.926 + 919G > A of the GLA gene) in the Taiwanese population [5,6,7,8]. Taking advantage of the nationwide database, we were able to pool a large number of Fabry disease cases from the population-based healthcare database.

The study has some limitations, and therefore, the findings should be interpreted with caution. First, despite using a nationwide healthcare dataset covering over 99.9% of Taiwanese residents, the number of Fabry disease patients could be underestimated due to the high prevalence rate of the pathogenic variant of cardiac Fabry disease (IVS4 + 919G > A) and the late-onset and indolent clinical course of Fabry disease. Therefore, there could be some misclassification bias with an indeterminate number of patients with Fabry disease misclassified as not having Fabry disease, causing the odds ratio to be biased toward 1.0, biasing results toward the null hypothesis. In addition, due to the deidentified nature of the dataset and patient privacy protections under the Personal Data Protection and related regulations of the NHIRD, it is not possible to validate the claims data by direct verification with patients. Furthermore, critical data such as laboratory data, genetic test results (of the GLA gene), audiometric data, Fabry disease severity score, the severity of hearing-related disorders or the laterality of hearing loss are not available. As such, further exploration of differences between cases and controls on the degree of hearing loss or tinnitus is not possible, as possible associations between the severity of auditory phenotypes and the severity of Fabry disease may be caused by different pathogenic GLA gene variants.

In conclusion, our study finds that Fabry disease is significantly associated with certain auditory manifestations, including hearing loss, sudden deafness, and tinnitus. Healthcare providers of patients with Fabry disease should be cautious about the possible otologic manifestations, and routine otologic consultation should be considered. Although epidemiologic evidence from this study suggests an association, studies among other racial groups and geographic regions are needed to further explore the observed associations.

## Figures and Tables

**Table 1 jcm-11-07396-t001:** Demographic characteristics of persons with Fabry disease and comparison patients (*n* = 13,872).

Variable	Patients with Fabry Disease (*n* = 2312)		Propensity Score-Matched Controls (*n* = 11,560)		*p* Value
	Total No.	%	Total No.	%	
Age, mean (SD)	50.1 (14.3)		50.1 (14.3)		0.990
Males	1039	44.9	5195	44.9	>0.999
Monthly Income					0.994
<NT$1~15,841	435	18.8	2167	18.8	
NT$15,841~25,000	929	40.2	4658	40.2	
≥NT$25,001	948	41.0	4735	41.0	
Geographic region					>0.999
Northern	1135	49.1	5675	49.1	
Central	295	12.8	1480	12.8	
Southern	805	34.8	4015	34.8	
Eastern	77	3.3	380	3.3	
Urbanization level					>0.999
1 (most urbanized)	647	28.0	3235	28.0	
2	787	34.0	3939	34.0	
3	418	18.1	2095	18.1	
4	231	10.0	1154	10.0	
5 (least urbanized)	229	9.9	1137	9.9	

**Table 2 jcm-11-07396-t002:** Prevalence rates of tinnitus, hearing loss and sudden deafness among patients with Fabry disease vs. comparison patients.

Presence of Tinnitus	Total (*n* = 13,872)		Patients with Fabry Disease (*n* = 2312)		Comparison Patients (*n* = 11,560)	
	*n*, %		*n*, %		*n*, %	
Yes	1742	12.6	385	16.7	1357	11.7
No	12,130	87.4	1927	83.3	10,203	88.3
Presence of hearing loss						
Yes	886	6.4	174	7.5	712	6.2
No	12,986	93.6	2138	92.5	10,848	93.8
Presence of sudden deafness						
Yes	156	1.1	39	1.7	117	1.0
No	13,716	98.9	2273	98.3	11,443	99.0

**Table 3 jcm-11-07396-t003:** Covariate-adjusted odds of tinnitus, hearing loss, and sudden deafness among Fabry disease vs. controls (*n* = 13,872).

Variable	*Odds Ratio (95% Confidence Interval, CIs)*		
	Tinnitus	Sudden Deafness	Hearing Loss
Fabry disease *(Non adjusted)*	1.502 *** (1.328~1.699)	1.678 ** (1.165~2.418)	1.240 * (1.044~1.473)
Fabry disease *(Adjusted ^a^)*	1.513 *** (1.336~1.713)	1.681 ** (1.166~2.423)	1.246 * (1.047~1.483)
Age	1.024 *** (1.020~1.027)	1.018 ** (1.007~1.030)	1.041 *** (1.035~1.046)
Sex	0.802 *** (0.722~0.891)	0.947 (0.704~1.347)	1.011 (0.878~1.165)
Monthly income			
<NT$15,841 (reference group)	1.000	1.000	1.000
NT$15,841~25,000	1.113 (0.964~1.285)	0.873 (0.558~1.365)	0.878 (0.727~1.061)
≥NT$25,001	1.119 (0.969~1.293)	1.081 (0.701~1.666)	0.977 (0.812~1.177)
Geographic region			
Northern (reference group)	1.000	1.000	1.000
Central	1.240 * (1.051~1.463)	0.733 (0.430~1.247)	1.027 (0.818~1.291)
Southern	1.128 * (1.006~1.265)	0.734 (0.506~1.063)	0.803 ** (0.684~0.943)
Eastern	0.809 (0.587~1.114)	0.611 (0.237~1.578)	0.910 (0.612~1.354)
Urbanization level			
1 (reference group)	1.000	1.000	1.000
2	1.005 (0.882~1.146)	1.518 (0.988~2.330)	1.044 (0.875~1.245)
3	0.992 (0.849~1.160)	1.173 (0.694~1.983)	0.905 (0.730~1.122)
4	0.966 (0.792~1.179)	2.734 *** (1.571~4.757)	1.023 (0.784~1.334)
5	1.051 (0.865~1.276)	0.880 (0.408~1.894)	0.794 (0.597~1.056)

Note: * *p* < 0.05; ** *p* < 0.01; *** *p* < 0.001; *^a^* Adjusted for age, sex, monthly income, geographic region and urbanization level.

## Data Availability

Data from the National Health Insurance Research Database, now managed by the Health and Welfare Data Science Center (HWDC), can be obtained by interested researchers through a formal application process addressed to the HWDC, Department of Statistics, Ministry of Health and Welfare, Taiwan (https://dep.mohw.gov.tw/DOS/lp-2506-113.html. accessed on 2 January 2022).

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
