# Peer review of "Association of Fabry Disease with Hearing Loss, Tinnitus, and Sudden Hearing Loss: A Nationwide Population-Based Study"

_jcm, 2022, doi:10.3390/jcm11247396_

Round 1

Reviewer 1 Report

Thanks for this work, the rationale of the work is very clear. However, I have some questions:

Why the authors did not describe the type, degree, and laterality of hearing loss?

Why the authors considered sudden HL as a sudden entity of the broad term of hearing loss??

How the authors judged the presence of HL: Is it based on patients’ descriptions or based on audiometric data?

Why the authors excluded cases with Fabry disease diagnosed before 2015??

Author Response

Thanks for this work, the rationale of the work is very clear. However, I have some questions:

Why the authors did not describe the type, degree, and laterality of hearing loss?

Response: Thanks for your suggestion. As stated in the Limitation, “Furthermore, critical data such as laboratory data, genetic test results (of the GLA gene), audiometric data, Fabry disease severity score, the severity of hearing-related disorders or laterality of hearing loss are not available.”

Why the authors considered sudden HL as a sudden entity of the broad term of hearing loss??

Response: Among all the hearing-related manifestations of Fabry disease, we think sudden hearing loss is the most emergent entity that warrants an urgent clinical visit and timely management, as delayed treatment may cause irreversible outcomes of hearing impairment. Vascular pathology is the leading cause of hearing-related disorders, including sudden hearing loss. As Fabry disease is regarded as vasculopathy, we think it may be highly associated with sudden hearing loss. We therefore conclude that healthcare providers of patients with Fabry disease should be cautious about the possible otologic manifestations, and routine otologic consultation should be considered.

How the authors judged the presence of HL: Is it based on patients’ descriptions or based on audiometric data?

Response: We identify the presence of HL by ICD diagnosis codes in the ambulatory visit claims, hearing loss (ICD-9-CM code 389 or ICD-10-CM codes H90 or H91). In Taiwan, if hearing loss is suspected, patient will receive audiometer tests from otologist, otolaryngologist, or ear, nose and throat doctor in order to make a definite medical diagnosis.

Why the authors excluded cases with Fabry disease diagnosed before 2015??

Response: Indeed, we did not exclude cases with Fabry disease diagnosed before 2015. As stated in Methods, “To identify patients in the study group, we extracted the claims records of all patients aged over 20 years with a diagnosis of Fabry disease (ICD-9-CM code 272.7 or ICD-10-CM code E75.21) in ambulatory care visits to clinics or outpatient departments of hospitals between January 1, 2015, and December 31, 2017.” After checking the dataset, we found that over 96% of patients with Fabry disease diagnosed before 2015 also receive Fabry diagnosis between 2015 and 2017.

Reviewer 2 Report

General comment : 

The article describes the prevalence of deafness, tinnitus, and the onset of sudden deafness in patients with Fabry disease. The interest of this article is to get data from a national database, with a control population, and with a correct statistical methodology. The discussion clearly presents the limitations of the study. 

Specific comment :

The article by Ciceran et al. from 2016 should be cited because it made a good summary

The histological description of the lesions presented on page 36 should be verified in the original article (mentions vacuolization of the endothelial cells? Sure?)

Author Response

General comment :

The article describes the prevalence of deafness, tinnitus, and the onset of sudden deafness in patients with Fabry disease. The interest of this article is to get data from a national database, with a control population, and with a correct statistical methodology. The discussion clearly presents the limitations of the study.

Specific comment :

The article by Ciceran et al. from 2016 should be cited because it made a good summary

Response: We have added this reference (reference 3) accordingly.

The histological description of the lesions presented on page 36 should be verified in the original article (mentions vacuolization of the endothelial cells? Sure?)

Response: In order to avoid the confusion, we have revised the relevant statements as follows: “They found cytoplasmic vacuolization of the vascular endothelial cells, hair cell loss in the basal turn of the cochlea, spiral ganglion loss, stria vascularis, and spiral ligament atrophy.”

Round 2

Reviewer 1 Report

I still have a major concern regard the missed audiometric data. However, I will accept the work in its form due to rarity f the disease. I suggested that authors should made a "complete" data base for this disease to be used in more comprehensive future work.